# Training outcomes for audiology students using virtual reality or traditional training methods

**David Bakhos**[1,2,3]*, **John Galvin**[3,4], **Jean-Marie Aoustin**[1,5], **Mathieu Robier**[1,5], **Sandrine Kerneis**[1], **Garance Bechet**[6], **Norbert Montembault**[6], **Stéphane Laurent**[6], **Benoit Godey**[6,7], **Charles Aussedat**[1,2]

**1** CHU de Tours, service ORL et Chirurgie Cervico-Faciale, Tours, France, **2** Faculté de Médecine de Tours, Université François-Rabelais de Tours, CHRU de Tours, Tours, France, **3** INSERM UMR 1253 I-brain, Université François-Rabelais de Tours, CHRU de Tours, Tours, France, **4** House Ear Institute, Los Angeles, California, United States of America, **5** Audilab, Saint-Pierre-des-Corps, France, **6** Ecole d'audioprothèse JE Bertin, Université de Rennes 1, Javene, France, **7** Service d'ORL et Chirurgie Cervico-Faciale, CHU de Rennes, Rennes, France

* david.bakhos@univ-tours.fr

**Data Availability Statement:** All relevant data are within the paper and its Supporting Information files.

## Abstract

Due to limited space and resources, it can be difficult to train students on audiological procedures adequately. In the present study, we compared audiology training outcomes between a traditional approach and a recently developed immersive virtual reality (VR) approach in audiology students. Twenty-nine first-year audiology students participated in the study; 14 received traditional training ("TT group"), and 15 received the VR training ("VRT group"). Pre- and post-training evaluation included a 20-item test developed by an audiology educator. Post-training satisfaction and self-confidence were evaluated using Likert scales. Mean post-training test scores improved by 6.9±9.8 percentage points in the TT group and by 21.1±7.8 points in the VRT group; the improvement in scores was significant for both groups. After completing the traditional training, the TT group was subsequently trained with the VR system, after which mean scores further improved by 7.5 points; there was no significant difference in post-VR training scores between the TT and VRT groups. After training, the TT and VRT groups completed satisfaction and self-confidence questionnaires. Satisfaction and self-confidence ratings were significantly higher for the VR training group, compared to the traditional training group. Satisfaction ratings were "good" (4 on Likert scale) for 74% of the TT group and 100% of the VRT group. Self-confidence ratings were "good" for 71% of the TT group and 92% of the VRT group. These results suggest that a VR training approach may be an effective alternative or supplement to traditional training for audiology students.

## Introduction

Training in audiometric procedures and diagnosis is an essential part of educating audiology students. Audiometric procedures include mathematical, physiological, and psychophysical rules. For example, when measuring audiometric thresholds, it is important to know the

**Funding:** Audilab provided support in the form of salaries for authors JMA, MR, NM but did not have any additional role in the study design, data collection and analysis, decision to publish, or preparation of the manuscript. The specific roles of these authors are articulated in the author contributions section.

**Competing interests:** JMA, MR, NM are employees of Audilab. There are no patents, products in development or marketed products to declare. This does not alter our adherence to PLOS ONE policies on sharing data and materials.

appropriate levels of contralateral noise masking to avoid misdiagnosis or, more importantly, overmasking. In France, during the first year of audiology education, audiometry training involves theoretical course work (80 hours), lectures, and clinical training. Traditional clinical training is performed using tutorials (3 hours), where students train with each other or with their instructor. During the second and third year, the students practice under the direct and indirect supervision of a licensed practitioner during their internship. Given the number of students, as well as limited resources in terms of the training time and space available, this supervision is not always possible. In addition, such training, even when supervised, does not meet the recommendation by the French National Authority for Health that students be sufficiently trained before they are allowed to work with patients or volunteers [1].

Simulations have become an increasing part of medical education. Using simulators has been shown to improve patient safety and reduce costs and morbidity [2]. Various computer-assisted simulators have been developed in recent years to support health professional training. These models have been created to improve the experience before medical students interact with human patients or volunteers. In this context, simulators are strongly supported by supervisors in terms of students' acquisition of both technical and non-technical skills [3]. Using simulators to train audiometry will better prepare students for future clinical practice. Over the last two decades, simulations have been increasingly integrated into medical education to facilitate the acquisition of knowledge and practice in a safe environment, allowing students to train and learn from their mistakes without risk to patients [4–7]. Simulators allow students to repeatedly learn and practice until they feel confident interacting with a patient. Currently, there are few simulators directed at developing students' audiometry skills.

One area of recent development is the use of virtual reality (VR) simulators, which allow for a more immersive experience for the learner. Given advances in computer technology, VR simulators for the medical field have been greatly improved. VR simulators have been shown to improve learning outcomes for various surgery training steps. VR simulators have also been used to train clinical reasoning in the field of traumatology to enhance decision-making among students [8]. We recently developed a VR simulator for audiometric training [9] that simulated the clinical environment. Seven clinical cases were simulated, including otosclerosis, presbycusis, vestibular schwannoma, incus luxation, malingering presentation, sudden idiopathic deafness, and bilateral hearing loss due to cholesteatoma. For each clinical case, "beginner" and "expert" modes were created. The beginner mode provides feedback during the training session to inform students of errors and to provide correction and explanations; expert mode provides no feedback. At the end of the VR training session, a report is produced that summarizes the errors made for each clinical case. External experts (medical educators and licensed practitioners) and otolaryngology students reported satisfactory validity [9].

The initial evaluation of the VR simulator involved expert educators and otolaryngology students that had already experienced traditional audiometry training. However, it is still unclear how training outcomes may differ between a traditional or VR simulation approach. We hypothesized that, given the differences between traditional and VR training, VR training would lead to equal or better post-training evaluation scores, compared to traditional training. We also expected greater student satisfaction with the VR training, given the immersive experience. The objective of this study was to compare training outcomes between a group of audiology students that received traditional audiometry training compared to a single session of VR training. We also evaluated students' satisfaction and self-confidence with the two training approaches.

## Methods

### Participants

We conducted a prospective single-center study with parallel groups at the Fougeres School of Audiology in France. Thirty-one first-year students were recruited for the study; two were excluded because they had repeated their first year. Accordingly, 29 students participated in the study (mean age = 21.1±4.1 years). The study was conducted in December 2019, after students had completed all theoretical course work for audiometry and audiometric findings in pathology. None of the participants had yet begun an internship at a hospital or audiology center. All data were anonymized prior to analysis. The ethics committee of Tours Hospital approved this project (2018–091). Students gave their verbal consent to participate to this study to a student in second year (BG) and their teacher (MN, LS).

Fourteen students (mean age = 20.4±1.7 years) were included in the Traditional Training (TT) group. The TT group received 3 hours of training supervised by a teacher in the audiology school. During the training, the teacher first reviewed basic audiometry principles; students were allowed to ask questions if they were not confident in their knowledge from the theorical lessons. Because the teacher had mild presbycusis, students were allowed to practice audiometry techniques for this sort of clinical case. The students were also allowed to train on each other. For example, a student would train with another student who simulated unilateral conductive hearing loss by plugging one ear with an ear plug.

Fifteen students (mean age = 21.7±5.6 years) were included in the Virtual Reality Training (VRT) group. The VR training was performed using the previously developed simulator [9]. During the training, a supervisor first explained how the VR system works and introduced the VR hardware, which included the headset, 2 captors, and handles (Oculus Rift ®). Next, each student was trained on 3 clinical cases (presbycusis, vestibular schwannoma, and sudden idiopathic deafness) using the beginner mode, which provided feedback during the session. Then, audiometric diagnosis and management were evaluated for each of the clinical cases, and a report was generated that summarized the errors during the evaluation. The duration of training for each case was approximately 30 minutes, and the total time of the VR training session was approximately 3 hours, including the 20 minutes of introducing the system and 20-minute breaks between clinical cases to avoid fatigue. Fig 1 shows screenshots of the VR system.

### Outcome measures

The evaluation was developed by experts in audiology teaching (i.e., authors MN, LS in the present study), and included 20 questions regarding otoscopy, pure-tone and speech audiometry, acoumetry, masking and tympanometry. The questions were based on the audiometry theoretical coursework and lectures; as such, the questions did not directly address the "hands on" practical training in the TT or VRT group. English and French versions of the evaluation can be found in S1 and S2 Appendices, respectively. The TT and VR training supervisors did not know the evaluation questions. The scoring for each question was weighted depending on the number of errors: 1 point in case of no error, 0.5 points in case of 1 error, 0.3 points in case of 2 errors, and no points in case of 3 or more errors. The maximum possible score was 20, and scores were converted to percent correct. The post-test was administered immediately after the training session for both groups. After completing the post-test evaluation, students in the TT group performed the same VR training as for the VRT group and then were re-tested.

After completing the TT or VR training and the post-test, post-training satisfaction and self-confidence were evaluated using subscales with 5-point Likert response (1 = "Strongly

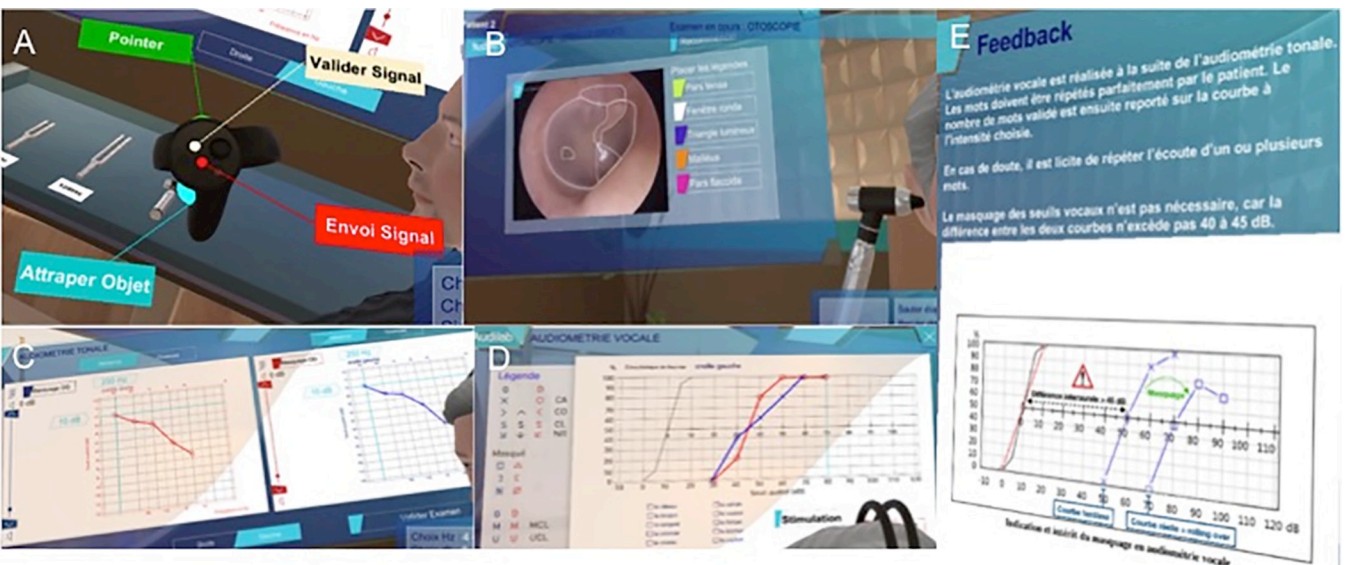

**Fig 1. Screenshots of the VR system for audiometric training.** A: Captions for the handle functions; B: Otoscopy interpretation; C: Pure-tone audiometry and determination of auditory thresholds; D: Speech audiometry and determination of auditory thresholds; E: Example of feedback regarding masking for speech audiometry.

disagree", 2 = "Disagree", 3 = "Indifferent", 4 = "Agree", 5 = "Strongly agree"). For satisfaction, the five items included: realism of training, feedback, support, comprehension, and degree of complement to the theoretical lessons. For self-confidence, the six items included: consultation, otoscopy interpretation, audiometry thresholds determination, speech audiometry procedures, masking, and overall confidence for clinical interaction with a patient. After training with the VR simulator, a four-item subscale with a 5-point Likert response was used to evaluate the immersive and realistic aspects of the VR simulator in relation to theoretical lessons using freely available online software (https://personalpages.manchester.ac.uk/staff/tim.wilding/PTA_Sim/index.html). Overall satisfaction or self-confidence ratings were calculated, and scores $\geq$ 4 were considered satisfactory.

## Statistical analyses

Statistical analyses was performed using GraphPad Prism V6 software (2002; version 8.0.0 for Windows; GraphPad Software Inc., San Diego, California USA: www.graphpad.com). Mann-Whitney U tests were used to compare training outcomes, training satisfaction, and self-confidence between groups. A Wilcoxon matched-pairs signed rank test was used to compare outcomes in the TT group with the traditional training or the subsequent VR training. Statistical significance was set at $p < 0.05$.

## Results

Fig 2 shows scores before and after training; mean scores are indicated by the horizontal lines. The mean pre-test score was 41.2±10.2 percent correct for the TT group and 39.5±6.5 percent correct for the VRT group. No significant differences in pre-test scores were observed between groups (U = 87; p = 0.444). The mean post-test score for the TT group significantly improved to 48.1±8.7 percent correct (W = 72; p = 0.021). Following the subsequent VR training, the mean TT post-test score further improved to 55.6±10.9 percent correct, significantly better

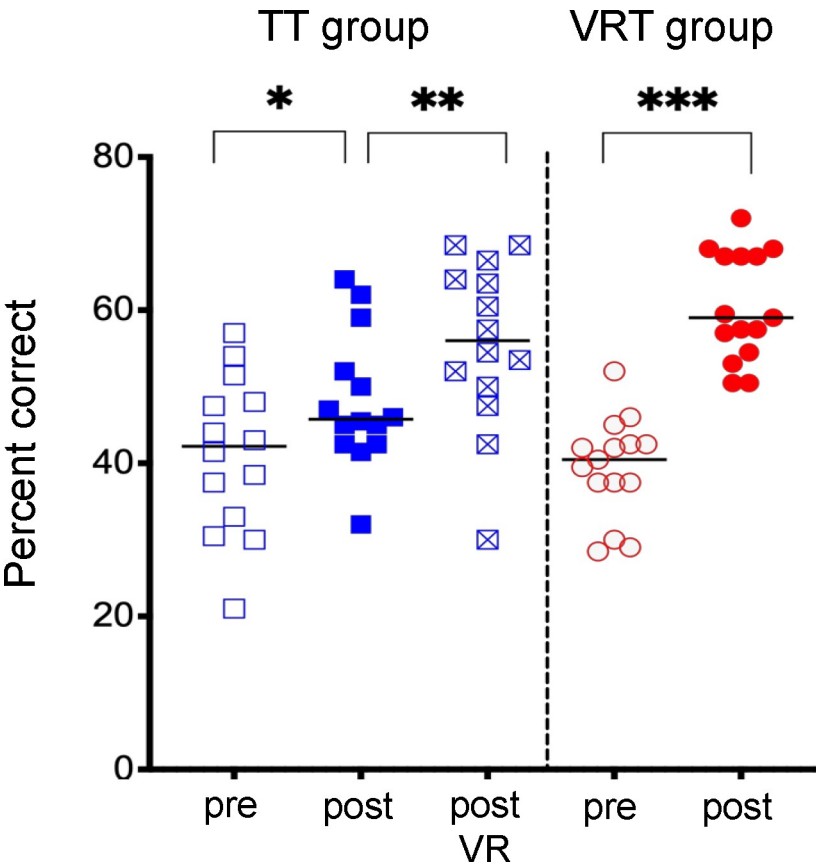

**Fig 2. Pre-test (open symbols) and post-test scores (filled symbols) for the TT group (blue) and the VRT group (red); data are also shown for the TT group after completing the VR training (blue squares with x).** The black horizontal lines show mean scores. The asterisks indicate significant differences (* = p < 0.05; ** = 0.001 < p < 0.05; *** = p < 0.001).

than pre-test scores (W = 95; p = 0.0012) and significantly better than post-test scores after the traditional training (W = 89; p = 0.0029). The mean post-test score for the VRT group significantly improved to 60.5±7.1 percent correct (W = 120; p<0.001). The mean improvement relative to pre-test scores was significantly larger for VRT group (21.1±7.8 percentage points) than for the TT group (6.9 ±9.8 percentage points) (U = 22.5; p<0.0001).

The post-training improvement in test scores was compared between the TT and VRT groups for individual questions. The post-training improvement was significantly larger for the VRT group than for the TT group for 4 questions: #6 (U = 44; p = 0.0046), #12 (U = 50.5; p = 0.0120), #13 (U = 54.5; p = 0.0117), and #17 (U = 49.5; p = 0.0117). These questions dealt with speech audiometry (#6), otoscopy (#12), tympanometry (#13) and audiometry interpretation of speech recognition thresholds (#17). For the remaining 16 questions, the post-training improvement remained significantly larger (U = 49.5; p = 0.014) for the VRT group (mean = 2.9±1.3 points) than for the TT group (mean = 1.29±1.47 points). After the subsequent VR training for the TT group, there was no significant difference in post-test scores between the TT group and the VRT group (U = 66; p = 0.0908).

Post-training satisfaction ratings were ≥ 4 ("Agree") in 74% of the TT group and 100% of the VRT group. Post-training self-confidence ratings were ≥ 4 in 71% of the TT group and 92% of the VRT group. Fig 3 shows the results for each item of the surveys. Significantly higher

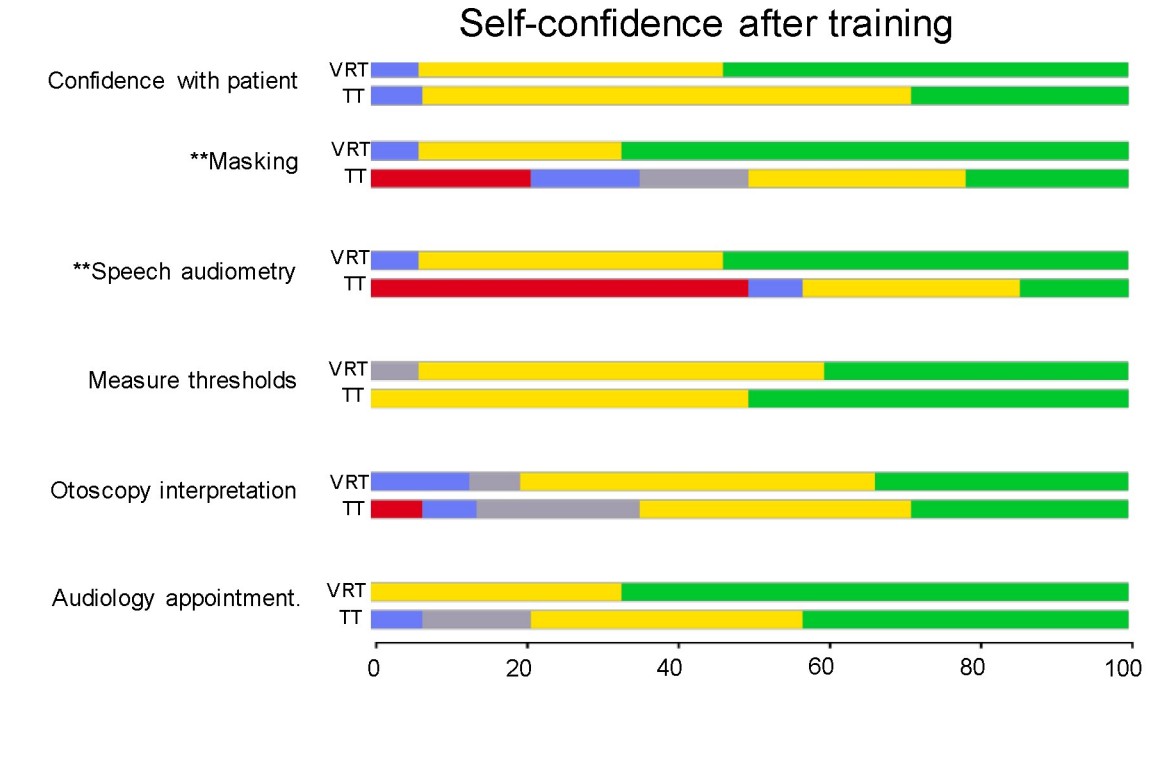

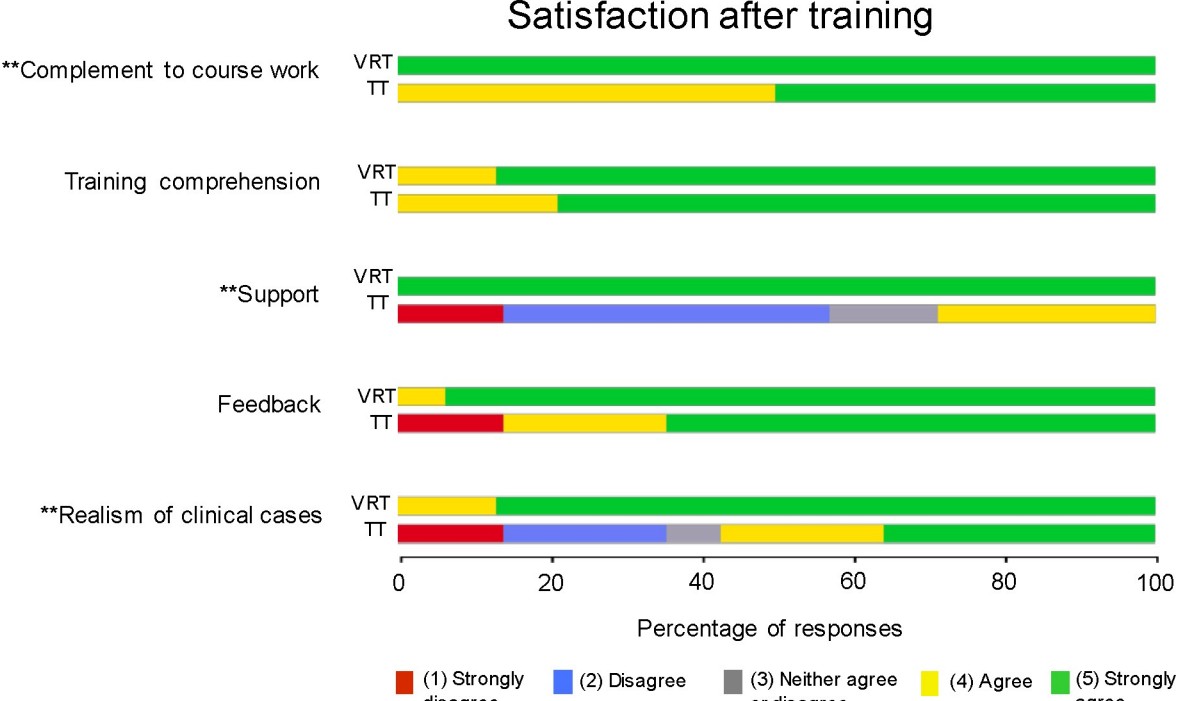

**Fig 3. Results of post-training satisfaction and self-confidence surveys for the VRT and TT groups.** For each item, data are shown for the VRT and TT groups. Each bar represents the percentage of respondents for each rating. The asterisks indicate significant differences between groups (*** = p<0.001).

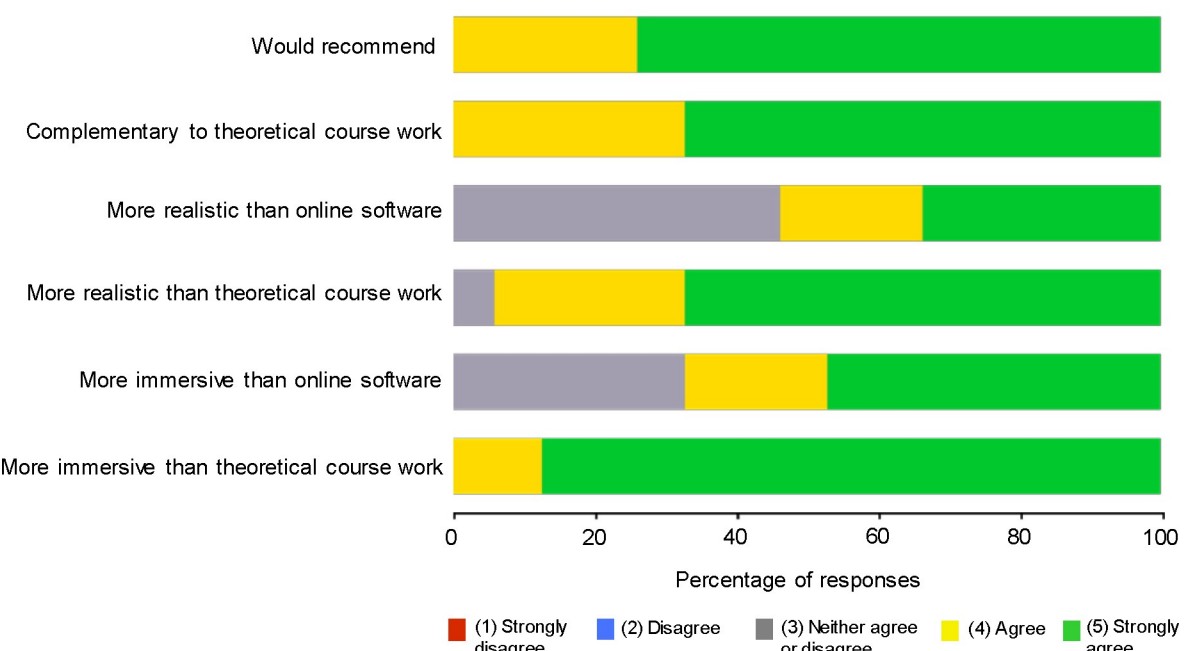

**Fig 4. Results of VRT quality surveys collected after VR training in the VR and TT groups.** Each bar represents the percentage of respondents for each rating.

ratings were observed in the VRT group than in the TT group for: realism of the audiological cases (U = 45.5; p = 0.0026), support during the training session (U = 0; p<0.0001), the degree of complement to the theoretical courses (U = 52.5; p = 0.0022), confidence for speech audiometry (U = 38.5; p = 0.0017), and confidence regarding the use of masking (U = 45; p = 0.0048).

Fig 4 shows the ratings for the VR system after the VRT and TT groups completed the VR training. Compared to theoretical courses, students agreed (a rating of 4) or strongly agreed (a rating of 5) that the VR training was more realistic (93%) and immersive (100%) than theoretical course work. They also agreed or strongly agreed that the VR training was more realistic (53%) and immersive (67%) than online software. All students (100%) agreed or strongly agreed that the VR training was complementary to theoretical courses and would recommend VR training to their colleagues.

## Discussion

The present study compared traditional audiometry training versus VR training for first-year audiology students. While post-test scores improved after training for both groups, the improvement in the VRT group was significantly larger than in the TT group. These findings suggest that the tested VR training may be a useful tool for audiology education. VR training offers the possibility for the students to practice their skills in an extensive and repetitive manner with feedback but without any consequences to real patients or volunteers [10]. We also found great enthusiasm for the VR training among the students tested.

Some previous studies have investigated interest in VR training versus traditional training in other medical fields. In general, VR training offered an advantage for learning theoretical knowledge over traditional, lecture-based education for dentistry [11] and anatomy [12]. In surgery, the goal of training is quite different because evaluation is based on the acquisition of technical skills. Some meta-analyses were conducted for gastro-intestinal endoscopy training [4, 5] and for surgery of the ear, nose, and throat [6] using VR. They showed that VR training alone allowed for similar acquisition of technical skills as with traditional training, and thus considered VR simulators to be a useful learning tool. As in our study, most previous studies that compared traditional and VR training underscore the students' enthusiasm for VR and the immersive environment. It would be ideal to simulate the same clinical cases for the traditional and VR training. However, this is more difficult for traditional training, where someone must "fake" some sort of hearing loss in the presence of their fellow students and instructor. The presentation of these simulated cases should also be consistent across students, which may be difficult. In the VR system, many more clinical cases can be easily and consistently implemented. This may partly explain the advantages of the VR training observed in the present study.

Feedback during training is essential to help students to improve performance, allowing students to learn from their mistakes quickly. In the present VR training system, concurrent feedback in beginner mode is cued by a sound alarm in case of error, followed by explanations of why the response was wrong. In this study, VR training was performed only using beginner mode. In our VR system (but not used in the present study), expert mode provides no concurrent feedback, but rather terminal feedback at the end of the evaluation. Some studies have suggested that terminal feedback is more effective than concurrent feedback, given the guidance hypothesis [13], which indicates that while concurrent feedback may help to learn a skill, learners are not dependent on concurrent feedback [14]. In the present study, students appreciated the concurrent feedback, with an overall satisfaction score of 4 for 80% of the TT group after the subsequent VR training, and for 100% of the VRT group. Only negative feedback was used in the VR training to indicate cases of error; positive feedback in cases of success has also demonstrated training benefits [15].

Traditional training for objective audiometry using normal-hearing volunteers, hearing-impaired volunteers, or manikin simulators has demonstrated improved self-confidence among audiology and speech language pathology students [16, 17]. This improved self-confidence has been also demonstrated after "boot-camps" [18], where students can practice with immediate feedback. Self-confidence as well as self-assessments play a major role in developing and improving clinical skills [19]. In the present study, the students similarly exhibited greater self-confidence after the training sessions, especially in the VRT group.

There are different types of simulators described in the literature, depending on their fidelity and resemblance to reality, divided into three main classes ranging from low- to high-fidelity [20]. Non-computerized manikins correspond to low-fidelity simulators (https://www.aheadsimulations.com/carl-for-training). They are used to train hearing aid manipulation, the removal of cerumen, real-ear unaided response measurements, and high-gain hearing aid fitting [21, 22]. Some computerized simulations allow training for specific tasks such as otoscopy, pure-tone air and bone conduction audiometry with online virtual patients [23]. Some of these online solutions are freely available (https://personalpages.manchester.ac.uk/staff/tim.wilding/PTA_Sim/index.html), while others require payments. One high-fidelity simulator exists in the field of audiology (Intelligent Hearing Systems, Miami, FL), and is a computerized manikin to train objective audiometry (e.g., auditory brainstem response, otoacoustic emissions). The present VR training software is somewhat different from these simulators, given that it is a 3D immersive experience, rather than some combination of hardware and software. Increasingly VR is considered to be a high-fidelity simulator [23, 24].

Some limitations to the present study should be discussed. First, our study was from a single center; a multi-center study is needed to confirm the present results. Second, we did not evaluate long-term retention of the training because students went on to internships at different audiology centers; it is possible that additional learning may have occurred at these centers, making it difficult to assess training retention. Previous studies have shown no difference in retention with traditional or VR training [25, 26]. However, greater satisfaction and engagement was reported by students who received VR training [26]. Similarly, greater satisfaction with VR training was reported in the present study, and students were very satisfied with the immersive and realistic environment. We note that some procedural learning may have been possible, because the same test was given before and after training (3 times for the TT group, as they were tested before training, after traditional training, and after the subsequent VR training). Even if some procedural learning may have occurred, the post-training improvement was more than double for VRT group, compared to the TT group. After traditional training, the mean improvement for the TT group was 6.9 percentage points; after completing the subsequent VR training, the mean score further improved by 7.5 percentage points. However, even after the VR training (and possible procedural learning), the mean score for the TT group remained 4.9 percentage points below that of the VRT group (although the difference at these endpoints was not significant). For future implementation, it would be advisable to create different questionnaires that cover different aspects of the clinical cases across pre- and post-training testing.

The present VR training system has also some limitations, especially for training situations in which precise hand movements are required (e.g., otoscopy, placement of air or bone transducers, using the audiometer, etc.). These gestures can be learned with a manikin or during a traditional training session with a teacher. Furthermore, the VR training does not presently permit evaluation of a student's ability to explain the audiometric procedure to a patient. However, that is the role of the hospital and audiological center internships during second and third year. Thus, the goal of VR is not to replace, but to complement other modes of audiological training.

The present VR simulator satisfies the recommendations by the French National Authority for Health, which requires sufficient training before students can work with patients or volunteers. In the future, the benefit of VR training may be greatly improved by including a broader range of clinical cases and procedures. For example, behavioral measures of auditory thresholds in children are challenging; future VR systems could train students on age-appropriate audiometric techniques (e.g., evaluating infant responses). Further studies could also be directed at norming VR evaluation scores across different levels of medical education. For example, it would be useful to know the range of scores for first-year and third-year students. Once this range is identified, the VR evaluation could also determine whether students have sufficiently progressed in their audiological education, or to know if VR offers similar benefits for basic or advanced training. Furthermore, a complementary study could be designed to investigate the benefits of feedback for traditional or VR training. For traditional training, feedback could be provided by a supervisor for each student; such an approach is likely to be time-consuming and might require multiple supervisors to provide feedback to all students. While probably resource-intensive, it would be interesting to compare the benefits of such feedback between traditional and VR training approaches. VR training could also be used for continuing education and certification programs for practitioners. Further studies are needed to demonstrate the real-life benefit of VR training for practitioners and patients and to determine whether VR training can reduce the number of medical errors in clinical practice.

## Conclusions

Simulations are widely used to train students in various fields of medicine; VR enables a safe training environment without risk to patients or volunteers. In the present study, our immersive VR training system provided audiology students with better learning outcomes and self-confidence than found with traditional training. Presently, the VR simulator can be used as a supplement to traditional audiology training; additional studies are needed to know whether it can replace traditional training. Further technological developments are also needed to expand the audiology training modules, such as behavioral and objective hearing tests for pediatric patients.

## Supporting information

**S1 Appendix.**
(DOCX)

**S2 Appendix.**
(DOCX)

**S1 Data.**
(XLSX)

## Acknowledgments

To the students and Fondation pour l'audition.

## Author Contributions

**Conceptualization:** David Bakhos, Charles Aussedat.

**Data curation:** Garance Bechet.

**Formal analysis:** David Bakhos, Sandrine Kerneis.

**Funding acquisition:** Jean-Marie Aoustin.

**Investigation:** Sandrine Kerneis, Garance Bechet, Stéphane Laurent, Benoit Godey.

**Methodology:** David Bakhos, Norbert Montembault, Stéphane Laurent, Benoit Godey.

**Project administration:** David Bakhos, Benoit Godey.

**Resources:** Mathieu Robier.

**Software:** Jean-Marie Aoustin, Mathieu Robier.

**Supervision:** David Bakhos, Stéphane Laurent, Benoit Godey.

**Validation:** David Bakhos, John Galvin, Jean-Marie Aoustin, Mathieu Robier, Sandrine Kerneis, Garance Bechet, Norbert Montembault, Stéphane Laurent, Benoit Godey, Charles Aussedat.

**Visualization:** David Bakhos, John Galvin, Charles Aussedat.

**Writing – original draft:** David Bakhos.

**Writing – review & editing:** John Galvin.

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
