## [Decision Letter · Decision Letter 0]

7 Sep 2020

PONE-D-20-21100

Virtual reality versus traditional audiometry training for audiology students

PLOS ONE

Dear Dr. Bakhos,

Thank you for submitting your manuscript to PLOS ONE. After careful consideration, we feel that it has merit but does not fully meet PLOS ONE’s publication criteria as it currently stands. Therefore, we invite you to submit a revised version of the manuscript that addresses the points raised during the review process.

We look forward to receiving your revised manuscript.

Kind regards,

Rafael da Costa Monsanto, M.D.

Academic Editor

PLOS ONE

Additional Editor Comments:

Dear authors,

The comments made by the reviewers are listed below. From a broad perspective, the subject is interesting and worthy of investigation. The data presented by the authors was mostly clear and congruent with the objectives.

However, as listed by the reviewers, there are still some concerns that need to be addressed before the manuscript is further considered for publication. Most relevant concerns:

1) Data availability: as per Plos One's "submission instructions", it is highly recommended that the authors deposit data either in an appropriate repository or included in a supporting information file.

2) English language: As pointed out by some of the reviewers, the english language should be critically revised.

3) Figure resolution: Fig. 3 should be improved.

4) Methodology: The reviewers pointed some information that lack further clarification, especially regarding potential significant differences between composition of groups.

Journal Requirements:

3. Please include additional information regarding the survey or questionnaire used in the study and ensure that you have provided sufficient details that others could replicate the analyses. For instance, if you developed a questionnaire as part of this study and it is not under a copyright more restrictive than CC-BY, please include a copy, in both the original language and English, as Supporting Information.  If the original language is written in non-Latin characters, for example Amharic, Chinese, or Korean, please use a file format that ensures these characters are visible.

4. Please state whether you validated the questionnaire prior to testing on study participants. Please provide details regarding the validation group within the methods section.

5. Thank you for stating the following in the Financial Disclosure section:

"Audilab funded the virtual reality system."

We note that one or more of the authors have an affiliation to the commercial funders of this research study : Audilab.

5.1. Please provide an amended Funding Statement declaring this commercial affiliation, as well as a statement regarding the Role of Funders in your study. If the funding organization did not play a role in the study design, data collection and analysis, decision to publish, or preparation of the manuscript and only provided financial support in the form of authors' salaries and/or research materials, please review your statements relating to the author contributions, and ensure you have specifically and accurately indicated the role(s) that these authors had in your study. You can update author roles in the Author Contributions section of the online submission form.

5.2. Please also provide an updated Competing Interests Statement declaring this commercial affiliation along with any other relevant declarations relating to employment, consultancy, patents, products in development, or marketed products, etc.  

6. We note that you have indicated that data from this study are available upon request. PLOS only allows data to be available upon request if there are legal or ethical restrictions on sharing data publicly. For information on unacceptable data access restrictions, please see http://journals.plos.org/plosone/s/data-availability#loc-unacceptable-data-access-restrictions.

Reviewers' comments:

Reviewer's Responses to Questions

**Comments to the Author**

1. Is the manuscript technically sound, and do the data support the conclusions?

Reviewer #1: Yes

Reviewer #2: Yes

Reviewer #3: Yes

Reviewer #4: Yes

2. Has the statistical analysis been performed appropriately and rigorously? 

Reviewer #1: Yes

Reviewer #2: Yes

Reviewer #3: N/A

Reviewer #4: I Don't Know

3. Have the authors made all data underlying the findings in their manuscript fully available?

Reviewer #1: Yes

Reviewer #2: Yes

Reviewer #3: Yes

Reviewer #4: No

4. Is the manuscript presented in an intelligible fashion and written in standard English?

Reviewer #1: No

Reviewer #2: Yes

Reviewer #3: Yes

Reviewer #4: Yes

5. Review Comments to the Author

Reviewer #1: COVID-19 has dramatically altered the audiology field at the patients and students. Application of the Virtual reality vs. traditional audiometry training for audiology students is highly needed as thus. So, thanks for the authors for their efforts.

The manuscript is well designed and written; however, it required an English editor. I made some suggestion in the attached file.

Good Luck!

Reviewer #2: Title: Virtual reality versus traditional audiometry training for audiology students

The manuscript focus on a very important issue, both clinically and scientifically.

The title is too broad and vague. Including “simulators” may direct the thoughts of the reader Virtual reality with case simulators versus traditional audiometry training for audiology students.

The hypothesis is that, since there is limited space and resources in most of the training hospitals, it can be difficult to adequately train students on audiological procedures in first-year audiology students.

The authors had already evaluated the VR simulator with expert educators and otolaryngology students that had already experienced traditional audiometry training. In this study, they compared training outcomes between a group of audiology students that received traditional audiometry training or a single session of VR training.

Introduction

Line 64 refers to a literature (8) after mentinioning (3)

“(3). Using simulators to train audiometry will better prepare students for future clinical 65 practice. Over the last two decades, simulations have been increasingly integrated into medical education to 66 facilitate acquisition of knowledge and practice in a safe environment, allowing students to train and learn from 67 their mistakes without risk to patients (8).” Would citations be (3-7)?

Methods

Both training sessions lasted around three hours. But they did not offer the same opportunities. Please specify.

How long after the trainings were the tests administered? Hours after the completion or days? Were this time equal for both groups?

The test was the same for both groups and is related to the theoretical subjects that were reinforced in the VR training that could not be covered in the TT. It would be interesting to highlight in the test, which questions covered those aspects. As for instance, question 6 covers word recognition score in vestibular schwannoma that was experienced in the VR training group, but not with the TT…. The fact that VR opens more case opportunities than TT, as well as the feedback, imposes a bias that the virtual reality itself not be the reason for the superior outcomes, but the opportunity of different case situations and feedback.

Tests used in the Statistical analysis were not mentioned in the Methods, although presented in the Results.

Results

Figure 3 is blurred, and captions are overlapped

Discussion

It is reasonable to aggree that the biggest advantage of VR is to offer this opportunity and feedback in a timely manner that TT would take probably the double time to offer the same case opportunities and one to one feedback.

The authors might explore the fact that the same opportunities could be given for both groups.

In the Discussion session, it would be interesting to add insights on to what extent do the authors think that VR involve. Do the authors think that virtual reality only applies to basic training?

Reviewer #3: The authors examined the effectiveness of a virtual-reality training in comparison to traditional training in first-year students of Audiology. This article seems to be a continuation of the authors previous article which was about the development of their VR system and its validation. The work seems interesting and note-worthy. The authors were aware of some limitations and honestly pointed out in Discussion. I have some criticism on this work.

- One big limitation of VR systems is their incapability to simulate situations in which hands are trained to for example proper otoscopy and placement of earphone or bone-vibrator and also communication with real people to take history and instruct them how to follow the test. How are these situations addressed in this VR system?

- Why students in VRT group did not test a real patient/person at the end in order to evaluate their confidence and satisfaction in a realistic manner besides answering the subscales?

- The method of statistical analysis of the data needs to be clarified.

Reviewer #4: Thank you for submitting this interesting and comprehensibly written manuscript. This work makes an important contribution to improve the audiometric training as part of audiological education. Some aspects of the manuscript should be clarified. The method section should be described more clearly, especially the use of feedback for the study participants. Also, information on statistics is missing and should be added.

Here are my specific comments below.

Abstract

Results: Please add if the improvements of the two groups were significantly different and if the results for satisfaction and self-confidence showed significant differences between the two groups.

Line 38: „scores additionally improved by 7.5 points.” rather than „scores improved by an additional 7.5 points. “

Line 42: “for audiology students” missing punctuation mark “for audiology students.”

Introduction

Line 50: “overstimulation”: Maybe “overmasking” is the better term.

Line 52: „with each other“ rather than „with each another“

Line 53: „with volunteers that have normal hearing or hearing loss“ better: “with normal hearing or hearing impaired volunteers”

Line 53: “second and third years” to “second and third year”

Line 62: remove “for the learner”

Line 72: “were greatly improved” rather than “have greatly improved”

Line 82: What exactly is the aim of the study, what is the hypothesis?

Participants and methods

Line 91: „(mean age: 21.1±4.1 years),“ wrong punctuation mark „(mean age: 21.1±4.1 years).“

Line 98: How were presbyacusis and unilateral deafness simulated by the teacher?

Line 99: „practice on each other“ missing punctuation mark „practice on each other.“

Line 103: “Next. each student” wrong punctuation mark “Next, each student”

Line 103 ff: Why were different hearing disorders simulated in both study groups? In the VRT group an additional vestibular schwannoma was simulated. Was the sudden idiopathic deafness in the VRT group also only unilateral, as in the TT group? The TT group has also practiced on normal hearing subjects (each other). Why wasn’t the same hearing impairment simulated in both groups? Does this possibly also have an effect on the learning outcome?

Line 104: Did the TT group also receive feedback like the VRT group (report and feedback during the measurements)?

Line 105: „Next, audiometry“ repetition, better „Then, audiometry“

Line 105: “audiometry diagnosis” better “audiometric diagnosis”

Line 106: Was the report given to the students? Did they receive the report as feedback? If so, has the TT group also received such report?

Line 107: How long did the measurements take in the TT group? Was the duration comparable to the VRT group?

Line 114: „tone and speech audiometry“ better „pure tone and speech audiometry“

Line 115: Is there a concrete reference to the practical exercises or can these questions be answered after the theory part, even without practical audiometry training?

Line 117: „in case or 3 or“ to „in case of 3 or“

Line 119: „After the completing“ better „After completing“

Line 127: “was also used” rather than “was also administered”

Line 128: What kind of online software is this? Is it used for education or is it generally freely available? Have all participants tested online software in advance?

Line 130: The study scheme is not quite clear. At what time which evaluation was made in which study group?

Line 130 ff: Was the assessment of self-confidence and satisfaction made before or after the training feedback? And was there feedback for both groups?

Line 130 ff: When did the TT group do the assessment of self-confidence and satisfaction, after the TT Training or after the subsequent VR training?

Was it possible to also simulate personal contact with the patient for the VRT group? (e.g. explanation of the measurement or measuring instructions)

Description of the statistics is missing. Which tests were used? Was the data checked for normal distribution?

Results

What do U and W stand for when specifying the statistics? Please state in the method section which statistical tests were used (presumably Wilcoxon and Man-Whitney-U?), then it is comprehensible.

Line 143: „posttest“ consistent to „post-test“

Line 150 to 153: Significant differences were found. Which group scored better in each case? Always the VRT group?

Line 157 ff: Was there a difference in the rating of the VR system between the two groups? The TT group has practically completed two trainings (TT and subsequent VR) and may have another perspective than the VRT, which „only“ did the VR training.

Discussion

Line 169-170: Why did the VRT group show so much better results than the TT? Are there any ideas that might explain that? Could the comprehensive feedback play a role at VRT Or could the different hearing disorders simulated in both groups have an influence on the learning outcome?

Line 176: „lecture-based education of“ rather than „lecture-based education teaching of“

Line 188 ff: Feedback seems to have an influence on the learning success. Has the TT group also received feedback? If so, to the same extent and concurrent or terminal? Could less or no feedback of the TT group explain the differences in the post-training outcomes between VRT and TT group?

Line 196-197: Could direct contact with patients be practiced in the VRT group, as was the case with real test subjects in the TT group? If not, could this influence the results regarding self-confidence?

Line 198-206: Where in this paragraph is the reference to the current study? To which type of simulation can the current study be assigned? What consequences does this have for the study results?

Line 210: „to assess training retention.“ rather than „to ascertain training retention.“

Line 214: “because the same test was given” rather than “given that the same test was given”

Line 214-215: The TT group even carried out the evaluation three times.

Line 218: „additionally improved by 7.5 percentage“ rather than „improved by an additional 7.5 percentage“

Limitations: Another limitation might be the fact that only the VRT group received feedback (If that was the case). This might influence the results, e.g. self-confidence.

Line 223: „directive“ better „guideline“?

Conclusions

Line 236: „VR enables a safe“ rather than „VR permits a safe“

Line 238: “with better learning outcomes” rather than “with better learning”

Figure legends

Figure 1, line 310: “Tone audiometry” better “Pure tone audiometry”

Figure 3: line 318: “VR” use “VRT”. Here, the abbreviation VR was used instead of VRT. Please use consistent in the manuscript, since VR is also used as a separate abbreviation.

Figure 4: line 322: „VR“ use „VRT“

Figure 3:

The text in the figure is not readable. The resolution is not good enough.

It is not easy to understand what is graphically represented here: Which group is the top bar, which is the bottom bar for the respective item? Please describe at least in the figure legend.

The graphic summarizes the results for both questionnaires (satisfaction and self-confidence). Please clearly indicate which questions belong to which questionnaire (e.g. separate them visually and label them accordingly in the figure legend).

6. PLOS authors have the option to publish the peer review history of their article (what does this mean?). If published, this will include your full peer review and any attached files.

Reviewer #1: No

Reviewer #2: No

Reviewer #3: **Yes: **Mansoureh Adel Ghahraman

Reviewer #4: No

---

## [Author Response · Author response to Decision Letter 0]

7 Nov 2020

To the editor:

We thank the reviewers and the editor for their helpful comments, and we have incorporated nearly all suggestions in the revised MS. Major changes include:

1. Greater detail in the Methods, including a section regarding statistical analyses.

2. Expanded Discussion section, especially regarding limitations of the present study.

3. Modified Figs 2-4 to improve resolution and include suggested figure details.

4. Careful proofreading to correct grammatical errors and typos, and to conform to PLOS One style.

We hope you find this revision acceptable. Let me know if you need further information.

Sincerely,

Pr David Bakhos, MD, PhD

Editors

We made the modifications

 � Add these informations

3. Please include additional information regarding the survey or questionnaire used in the study and ensure that you have provided sufficient details that others could replicate the analyses. For instance, if you developed a questionnaire as part of this study and it is not under a copyright more restrictive than CC-BY, please include a copy, in both the original language and English, as Supporting Information. If the original language is written in non-Latin characters, for example Amharic, Chinese, or Korean, please use a file format that ensures these characters are visible.

 � We add it 

4. Please state whether you validated the questionnaire prior to testing on study participants. Please provide details regarding the validation group within the methods section.

 � These evaluation was done by the teachers, we add this information

5. Thank you for stating the following in the Financial Disclosure section:

"Audilab funded the virtual reality system."

We note that one or more of the authors have an affiliation to the commercial funders of this research study : Audilab.

 � We made the modifications on line and in the cover letter 

6. We note that you have indicated that data from this study are available upon request. PLOS only allows data to be available upon request if there are legal or ethical restrictions on sharing data publicly. For information on unacceptable data access restrictions, please see http://journals.plos.org/plosone/s/data-availability#loc-unacceptable-data-access-restrictions.

 � We add a xls file with all the data

We made the modifications

Reviewer #1: 

COVID-19 has dramatically altered the audiology field at the patients and students. Application of the Virtual reality vs. traditional audiometry training for audiology students is highly needed as thus. So, thanks for the authors for their efforts. The manuscript is well designed and written; however, it required an English editor. I made some suggestion in the attached file.

Thank you for your comments and suggestions in the text, we really appreciate it. The second author from the US has also contributed some edits to the language to the revision.

Reviewer #2: 

The manuscript focus on a very important issue, both clinically and scientifically.

The title is too broad and vague. Including “simulators” may direct the thoughts of the reader Virtual reality with case simulators versus traditional audiometry training for audiology students.

The title has been changed to: “Training outcomes for audiology students using virtual reality or traditional training methods”

Introduction

Line 64 refers to a literature (8) after mentioning (3)

“(3). Using simulators to train audiometry will better prepare students for future clinical practice. Over the last two decades, simulations have been increasingly integrated into medical education to facilitate acquisition of knowledge and practice in a safe environment, allowing students to train and learn from their mistakes without risk to patients (8).” Would citations be (3-7)?

We have corrected the order of citations throughout

Methods

Both training sessions lasted around three hours. But they did not offer the same opportunities. Please specify.

We have modified this section of the Methods: “Fourteen students (mean age = 20.4±1.7 years) were included in the Traditional Training (TT) group. The TT group received 3 hours of training supervised by a teacher in the audiology school. During the training, the teacher first reviewed basic audiometry principles; students were allowed to ask questions if they were not confident in their knowledge from the theorical lessons. Because the teacher had mild presbycusis, students were allowed to practice audiometry techniques for this sort of clinical case. The students were also allowed to train on each other. For example, a student would train with another student who simulated unilateral conductive hearing loss by plugging one ear with an ear plug. 

Fifteen students (mean age = 21.7±5.6 years) were included in the Virtual Reality Training (VRT) group. The VR training was performed using the previously developed simulator (9). During the training, a supervisor first explained how the VR system works and introduced the VR hardware, which included the headset, 2 captors, and handles (Oculus Rift ®). Next, each student was trained on 3 clinical cases (presbycusis, vestibular schwannoma, and sudden idiopathic deafness) using the beginner mode, which provided feedback during the session. Then, audiometric diagnosis and management were evaluated for each of the clinical cases, and a report was generated that summarized the errors during the evaluation. The duration of training for each case was approximately 30 minutes, and the total time of the VR training session was approximately 3 hours, including the 20 minutes of introducing the system and 20-minute breaks between clinical cases to avoid fatigue. Fig 1 shows screenshots of the VR system.”

How long after the trainings were the tests administered? Hours after the completion or days? Were this time equal for both groups?

We have added: “The post-test was performed immediately after the session training for both groups.”

The test was the same for both groups and is related to the theoretical subjects that were reinforced in the VR training that could not be covered in the TT. It would be interesting to highlight in the test, which questions covered those aspects. As for instance, question 6 covers word recognition score in vestibular schwannoma that was experienced in the VR training group, but not with the TT…. The fact that VR opens more case opportunities than TT, as well as the feedback, imposes a bias that the virtual reality itself not be the reason for the superior outcomes, but the opportunity of different case situations and feedback.

We have added to the new Outcome Measures section of the Methods: “The evaluation was developed by experts in audiology teaching (i.e., authors MN, LS in the present study), and included 20 questions regarding otoscopy, pure-tone and speech audiometry, acoumetry, masking and tympanometry. The questions were based on the audiometry theoretical coursework and lectures; as such, the questions did not directly address the “hands on” practical training in the TT or VRT group. English and French versions of the evaluation can be found in Appendices 1 and 2, respectively.”

Tests used in the Statistical analysis were not mentioned in the Methods, although presented in the Results.

We have added a section “Statistical analyses” to the Methods. 

Results

Figure 3 is blurred, and captions are overlapped

We have improved Fig 3 resolution 

Discussion

It is reasonable to agree that the biggest advantage of VR is to offer this opportunity and feedback in a timely manner that TT would take probably the double time to offer the same case opportunities and one to one feedback. The authors might explore the fact that the same opportunities could be given for both groups.

We have added to the Discussion: “Furthermore, a complementary study could be designed to investigate the benefits of feedback for traditional or VR training. For traditional training, feedback could be provided by a supervisor for each student; such an approach is likely to be time-consuming and might require multiple supervisors to provide feedback to all students. While probably resource-intensive, it would be interesting to compare the benefits of such feedback between traditional and VR training approaches.”

In the Discussion session, it would be interesting to add insights on to what extent do the authors think that VR involve. Do the authors think that virtual reality only applies to basic training?

We have added to the Discussion: “Further studies could also be directed at norming VR evaluation scores across different levels of medical education. For example, it would be useful to know the range of scores for first-year and third-year students. Once this range is identified, the VR evaluation could also determine whether students have sufficiently progressed in their audiological education, or to know if VR offers similar benefits for basic or advanced training.”

Reviewer #3: 

- One big limitation of VR systems is their incapability to simulate situations in which hands are trained to for example proper otoscopy and placement of earphone or bone-vibrator and also communication with real people to take history and instruct them how to follow the test. How are these situations addressed in this VR system?

We have added: “The present VR training system has also some limitations, especially for training situations in which precise hand movements are required (e.g., otoscopy, placement of air or bone transducers, using the audiometer, etc.). These gestures can be learned with a manikin or during a traditional training session with a teacher. Furthermore, the VR training does not presently permit evaluation of a student’s ability to explain the audiometric procedure to a patient. However, that is the role of the hospital and audiological center internships during second and third year. Thus, the goal of VR is not to replace, but to complement other modes of audiological training.”

- Why students in VRT group did not test a real patient/person at the end in order to evaluate their confidence and satisfaction in a realistic manner besides answering the subscales?

As stated in the Introduction and Discussion, students are not allowed to work with real patients or volunteers until they have demonstrated sufficient competence after training, according to the recommendations by the French National Authority for Health.

- The method of statistical analysis of the data needs to be clarified.

We have added a section “Statistical analyses” to the Methods. 

Reviewer #4: 

The method section should be described more clearly, especially the use of feedback for the study participants. Also, information on statistics is missing and should be added.

We have added information to the Methods, as well as new section “Statistical analyses.”

Here are my specific comments below.

Abstract

Results: Please add if the improvements of the two groups were significantly different and if the results for satisfaction and self-confidence showed significant differences between the two groups.

We have added statistical significance to the evaluation and rating data.

Line 38: „scores additionally improved by 7.5 points.” rather than „scores improved by an additional 7.5 points.“

Revised as: “After completing the traditional training, students were trained with the VR system, after which mean scores further improved by 7.5 points.”

Line 42: “for audiology students” missing punctuation mark “for audiology students.”

Introduction

Corrected.

Line 50: “overstimulation”: Maybe “overmasking” is the better term.

Changed as suggested.

Line 52: „with each other“ rather than „with each another“

Corrected.

Line 53: „with volunteers that have normal hearing or hearing loss“ better: “with normal hearing or hearing impaired volunteers”

Changed as suggested.

Line 53: “second and third years” to “second and third year”

Corrected.

Line 62: remove “for the learner”

Changed as suggested.

Line 72: “were greatly improved” rather than “have greatly improved”

Revised as: “Given advances in computer technology, VR simulators for the medical field have been greatly improved.”

Line 82: What exactly is the aim of the study, what is the hypothesis?

We have modified the last paragraph of the Introduction: “The initial evaluation of the VR simulator involved expert educators and otolaryngology students that had already experienced traditional audiometry training. However, it is still unclear how training outcomes may differ between a traditional or VR simulation approach. We hypothesized that, given the differences between traditional and VR training, VR training would lead to equal or better post-training evaluation scores, compared to traditional training. We also expected greater student satisfaction with the VR training, given the immersive experience. The objective of this study was to compare training outcomes between a group of audiology students that received traditional audiometry training compared to a single session of VR training. We also evaluated students’ satisfaction and self-confidence with the two training approaches.”

Participants and methods

Line 91: „(mean age: 21.1±4.1 years),“ wrong punctuation mark „(mean age: 21.1±4.1 years).“

Changed to: “…(mean age = 21.1±4.1 years).”

Line 98: How were presbyacusis and unilateral deafness simulated by the teacher?

We have clarified: “During the training, the teacher first reviewed basic audiometry principles; students were allowed to ask questions if they were not confident in their knowledge from the theorical lessons. Because the teacher had mild presbycusis, students were allowed to practice audiometry techniques for this sort of clinical case. The students were also allowed to train on each other. For example, a student would train with another student who simulated unilateral conductive hearing loss by plugging one ear with an ear plug.”

Line 99: „practice on each other“ missing punctuation mark „practice on each other.“

Corrected.

Line 103: “Next. each student” wrong punctuation mark “Next, each student”

Corrected.

Line 103 ff: Why were different hearing disorders simulated in both study groups? In the VRT group an additional vestibular schwannoma was simulated. Was the sudden idiopathic deafness in the VRT group also only unilateral, as in the TT group? The TT group has also practiced on normal hearing subjects (each other). Why wasn’t the same hearing impairment simulated in both groups? Does this possibly also have an effect on the learning outcome?

It is true that “hands on” training was provided for only 2 clinical cases in the TT group, and 3 different cases in the VRT group. However, the evaluation was based on the theoretical learning rather than these specific cases. We have added: “The questions were based on the audiometry theoretical coursework and lectures; as such, the questions did not directly address the “hands on” practical training in the TT or VRT group. English and French versions of the evaluation can be found in Appendices 1 and 2, respectively. The TT and VR training supervisors did not know the evaluation questions.”

Line 104: Did the TT group also receive feedback like the VRT group (report and feedback during the measurements)?

In the TT group, there was some feedback provided by the instructor as needed. In the VRT group, feedback was provided continuously during the training.

Line 105: „Next, audiometry“ repetition, better „Then, audiometry“

Changed as suggested.

Line 105: “audiometry diagnosis” better “audiometric diagnosis”

Changed as suggested.

Line 106: Was the report given to the students? Did they receive the report as feedback? If so, has the TT group also received such report?

It is true that the VRT group received case-specific reports after training. There was no such report for the TT group. However, both groups received the evaluation test scores and were able to discuss the results with the instructor, which is some sort of common feedback across the groups.

Line 107: How long did the measurements take in the TT group? Was the duration comparable to the VRT group?

As we report in the revised MS, both groups received approximately 3 hours of training.

Line 114: „tone and speech audiometry“ better „pure tone and speech audiometry“

Changed as: “…pure-tone and speech audiometry…’

Line 115: Is there a concrete reference to the practical exercises or can these questions be answered after the theory part, even without practical audiometry training?

It’s true that the evaluation could be administered without any practical training. However, the practical training is meant to reinforce the theoretical training, and to increase confidence in clinical practice.

Line 117: „in case or 3 or“ to „in case of 3 or“

Sentence revised as: “The scoring for each question was weighted depending on the number of errors: 1 point in case of no error, 0.5 points in case of 1 error, 0.3 points in case of 2 errors, and no points in case of 3 or more errors.”

Line 119: „After the completing“ better „After completing“

Sentence revised as: “After completing the post-test evaluation, students in the TT group performed the same VR training as for the VRT group and then were re-tested.”

Line 127: “was also used” rather than “was also administered”

Line 128: What kind of online software is this? Is it used for education or is it generally freely available? Have all participants tested online software in advance?

In response to both comments, sentence revised as: “After training with the VR simulator, a four-item subscale with a 5-point Likert response was used to evaluate the immersive and realistic aspects of the VR simulator in relation to theoretical lessons using freely available online software (https://personalpages.manchester.ac.uk/staff/tim.wilding/PTA_Sim/index.html).”

Line 130: The study scheme is not quite clear. At what time which evaluation was made in which study group?

Line 130 ff: Was the assessment of self-confidence and satisfaction made before or after the training feedback? And was there feedback for both groups?

Line 130 ff: When did the TT group do the assessment of self-confidence and satisfaction, after the TT Training or after the subsequent VR training?

In response to all comments, we have clarified in the Methods: “The post-test was administered immediately after the training session for both groups. After completing the post-test evaluation, students in the TT group performed the same VR training as for the VRT group and then were re-tested.” Also see above responses.

Was it possible to also simulate personal contact with the patient for the VRT group? (e.g. explanation of the measurement or measuring instructions)

This function has not yet been developed for the present VR training system, but would be an important feature for future development. For now, we have added to the Discussion: “Furthermore, the VR training does not presently permit evaluation of a student’s ability to explain the audiometric procedure to a patient. However, that is the role of the hospital and audiological center internships during second and third year. Thus, the goal of VR is not to replace, but to complement other modes of audiological training.”

Description of the statistics is missing. Which tests were used? Was the data checked for normal distribution?

What do U and W stand for when specifying the statistics? Please state in the method section which statistical tests were used (presumably Wilcoxon and Man-Whitney-U?), then it is comprehensible.

In response to both comments, we have added a section “Statistical methods” to the Methods. 

Results

Line 143: „posttest“ consistent to „post-test“

We have changed to post-test throughout.

Line 150 to 153: Significant differences were found. Which group scored better in each case? Always the VRT group?

Revised as: “Significantly higher ratings were observed in the VRT group than in the TT group for: realism of the audiological cases (U=45.5; p=0.0026), support during the training session (U=0; p<0.0001), the degree of complement to the theoretical courses (U=52.5; p=0.0022), confidence for speech audiometry (U=38.5; p=0.0017), and confidence regarding the use of masking (U=45; p=0.0048).”

Line 157 ff: Was there a difference in the rating of the VR system between the two groups? The TT group has practically completed two trainings (TT and subsequent VR) and may have another perspective than the VRT, which „only“ did the VR training.

We have added: “After the subsequent VR training for the TT group, there was no significant difference in post-test scores between the TT group and the VRT group (U= 66; p=0.0908).”

Discussion

Line 169-170: Why did the VRT group show so much better results than the TT? Are there any ideas that might explain that? Could the comprehensive feedback play a role at VRT Or could the different hearing disorders simulated in both groups have an influence on the learning outcome?

We have added: “Some previous studies have investigated interest in VR training versus traditional training in other medical fields. In general, VR training offered an advantage for learning theoretical knowledge over traditional, lecture-based education for dentistry (11) and anatomy (12). In surgery, the goal of training is quite different because evaluation is based on the acquisition of technical skills. Some meta-analyses were conducted for gastro-intestinal endoscopy training (4, 5) and for surgery of the ear, nose, and throat (6) using VR. They showed that VR training alone allowed for similar acquisition of technical skills as with traditional training, and thus considered VR simulators to be a useful learning tool. As in our study, most previous studies that compared traditional and VR training underscore the students’ enthusiasm for VR and the immersive environment. It would be ideal to simulate the same clinical cases for the traditional and VR training. However, this is more difficult for traditional training, where someone must “fake” some sort of hearing loss in the presence of their fellow students and instructor. The presentation of these simulated cases should also be consistent across students, which may be difficult. In the VR system, many more clinical cases can be easily and consistently implemented. This may partly explain the advantages of the VR training observed in the present study.”

Line 176: „lecture-based education of“ rather than „lecture-based education teaching of“

Revised as: “In general, VR training offered an advantage for learning theoretical knowledge over traditional, lecture-based education for dentistry (11) and anatomy (12).”

Line 188 ff: Feedback seems to have an influence on the learning success. Has the TT group also received feedback? If so, to the same extent and concurrent or terminal? Could less or no feedback of the TT group explain the differences in the post-training outcomes between VRT and TT group?

We have added near the end of the Discussion: “Furthermore, a complementary study could be designed to investigate the benefits of feedback for traditional or VR training. For traditional training, feedback could be provided by a supervisor for each student; such an approach is likely to be time-consuming and might require multiple supervisors to provide feedback to all students. While probably resource-intensive, it would be interesting to compare the benefits of such feedback between traditional and VR training approaches.”

Line 196-197: Could direct contact with patients be practiced in the VRT group, as was the case with real test subjects in the TT group? If not, could this influence the results regarding self-confidence?

As pointed out in the revised Introduction: “Traditional clinical training is performed using tutorials (3 hours), where students will train with each other or with their instructor. During the second and third year, the students will practice under the direct and indirect supervision of a licensed practitioner during their internship. Given the number of students, as well as limited resources in terms of the training time and space available, this supervision is not always possible. In addition, such training, even when supervised, does not meet the recommendation by the French National Authority for Health that students be sufficiently trained before they are allowed to work with patients or volunteers (1).”

Line 198-206: Where in this paragraph is the reference to the current study? To which type of simulation can the current study be assigned? What consequences does this have for the study results?

We have added: “The present VR training software is somewhat different from these simulators, given that it is a 3D immersive experience, rather than some combination of hardware and software. Increasingly VR is considered to be a high-fidelity simulator.”

Line 210: „to assess training retention.“ rather than „to ascertain training retention.“

Changed as suggested.

Line 214: “because the same test was given” rather than “given that the same test was given”

Changed as suggested.

Line 214-215: The TT group even carried out the evaluation three times.

Revised as: “We note that some procedural learning may have been possible, because the same test was given before and after training (3 times for the TT group, as they were tested before training, after traditional training, and after the subsequent VR training).”

Line 218: „additionally improved by 7.5 percentage“ rather than „improved by an additional 7.5 percentage“

Revised as: “After traditional training, the mean improvement for the TT group was 6.9 percentage points; after completing the subsequent VR training, the mean score further improved by 7.5 percentage points.”

Limitations: Another limitation might be the fact that only the VRT group received feedback (If that was the case). This might influence the results, e.g. self-confidence.

We have added to the Discussion: “Furthermore, a complementary study could be designed to investigate the benefits of feedback for traditional or VR training. For traditional training, feedback could be provided by a supervisor for each student; such an approach is likely to be time-consuming and might require multiple supervisors to provide feedback to all students. While probably resource-intensive, it would be interesting to compare the benefits of such feedback between traditional and VR training.”

Line 223: „directive“ better „guideline“?

Revised as: “The current VR simulator satisfies the recommendations by the French National Authority for Health, which requires sufficient training before students can work with patients or volunteers.”

Conclusions

Line 236: „VR enables a safe“ rather than „VR permits a safe“

Line 238: “with better learning outcomes” rather than “with better learning”

In response to both comments, we have revised as: “Simulations are widely used to train students in various fields of medicine; VR enables a safe training environment without risk to patients or volunteers. In the present study, our immersive VR training system provided audiology students with better learning outcomes and self-confidence than found with traditional training.”

Figure legends

Figure 1, line 310: “Tone audiometry” better “Pure tone audiometry”

Changed as “…Pure-tone audiometry…”

Figure 3: line 318: “VR” use “VRT”. Here, the abbreviation VR was used instead of VRT. Please use consistent in the manuscript, since VR is also used as a separate abbreviation.

Figure 4: line 322: „VR“ use „VRT“

In response to both comments, changed as suggested.

Figure 3:

The text in the figure is not readable. The resolution is not good enough.

It is not easy to understand what is graphically represented here: Which group is the top bar, which is the bottom bar for the respective item? Please describe at least in the figure legend.

The graphic summarizes the results for both questionnaires (satisfaction and self-confidence). Please clearly indicate which questions belong to which questionnaire (e.g. separate them visually and label them accordingly in the figure legend).

We have modified and improved the resolution of Figs 2-4, and added the suggested details to the Fig. 3 caption.

---

## [Decision Letter · Decision Letter 1]

20 Nov 2020

Training outcomes for audiology students using virtual reality or traditional training methods

PONE-D-20-21100R1

Dear Dr. Bakhos,

We’re pleased to inform you that your manuscript has been judged scientifically suitable for publication and will be formally accepted for publication once it meets all outstanding technical requirements.

Kind regards,

Rafael da Costa Monsanto, M.D.

Academic Editor

PLOS ONE

Additional Editor Comments (optional):

Reviewers' comments:

Reviewer's Responses to Questions

**Comments to the Author**

1. If the authors have adequately addressed your comments raised in a previous round of review and you feel that this manuscript is now acceptable for publication, you may indicate that here to bypass the “Comments to the Author” section, enter your conflict of interest statement in the “Confidential to Editor” section, and submit your "Accept" recommendation.

Reviewer #1: All comments have been addressed

Reviewer #3: All comments have been addressed

Reviewer #4: All comments have been addressed

2. Is the manuscript technically sound, and do the data support the conclusions?

Reviewer #1: Yes

Reviewer #3: Yes

Reviewer #4: Yes

3. Has the statistical analysis been performed appropriately and rigorously? 

Reviewer #1: Yes

Reviewer #3: Yes

Reviewer #4: Yes

4. Have the authors made all data underlying the findings in their manuscript fully available?

Reviewer #1: Yes

Reviewer #3: Yes

Reviewer #4: Yes

5. Is the manuscript presented in an intelligible fashion and written in standard English?

Reviewer #1: Yes

Reviewer #3: Yes

Reviewer #4: Yes

6. Review Comments to the Author

Reviewer #1: (No Response)

Reviewer #3: (No Response)

Reviewer #4: Dear author,

thank you for this interesting manuscript. I have no more comments. All of my questions and comments have been answered and considered very well.

Kind regards.

7. PLOS authors have the option to publish the peer review history of their article (what does this mean?). If published, this will include your full peer review and any attached files.

Reviewer #1: **Yes: **RAZAN AL FAKIR

Reviewer #3: No

Reviewer #4: No

---

## [Editor Report · Acceptance letter]

25 Nov 2020

PONE-D-20-21100R1 

Training outcomes for audiology students using virtual reality or traditional training methods 

Dear Dr. Bakhos:

I'm pleased to inform you that your manuscript has been deemed suitable for publication in PLOS ONE. Congratulations! Your manuscript is now with our production department. 

Kind regards, 

on behalf of

Dr. Rafael da Costa Monsanto 

Academic Editor

PLOS ONE